# OpenReview forum: "Omni-Modal Large Language Models Jailbreaking with Adaptive Agent"
_ICLR.cc/2026/Conference — ICLR 2026 Conference Withdrawn Submission_

### Official Review · Reviewer_XeGS · 2025-10-30

**Soundness:** 3
**Presentation:** 1
**Contribution:** 2
**Rating:** 4
**Confidence:** 3

**Summary:**

This paper proposes a new framework, AdvOmniAgent, for jailbreaking Omni-MLLMs. To address the gradient shattering problem, this paper utilizes a two-stage optimization to
perform semantic-level updates for multimodal queries. After iterative refinement, the method could jailbreak the Omni-MLLMs successfully.

While the technical ideas are interesting, the paper's contributions are obscured by poor writing.

**Strengths:**

1. This paper proposes a new framework for jailbreaking Omni-MLLMs, contributing to the fields.
2. Comprehensive experiments are conducted to thoroughly demonstrate the effectiveness of the proposed method.
3. This paper formulates the optimization process as a search for the optimal input, which is realized by leveraging the LLM itself. It seems interesting.

**Weaknesses:**

1. The writing format should be improved:
    - The citation formatting is inconsistent throughout the paper — some references are preceded by a space while others are not. I recommend adding a space before all citations (e.g., “method [5]”), which is a basic academic formatting standard.
    - There are typographical errors in the use of quotation marks in the LaTeX source. The authors should correct the quotation style (use ``text'' instead of "text") for proper LaTeX formatting. The authors should revise all the use of quotations. For example, lines 110, 111, 112, 122.
    - Missing space after punctuation (e.g., line 116). Please add a space before the next sentence for correct formatting.
    - All notations should be clear. In the paragraph starting at line 120, the authors define $q^{aud}$ and $q^{text}$, but why don't give any definition of $q^{vis}$? Furthermore, the authors have no definition of $\mathcal{L}_{jb}$ across the whole paper.
    - All notations should be consistent. For example, `tri-modal` and `trimodal` are used interchangeably throughout the whole paper, which is not professional.
    - In line 409: `As shown in FiguresFigure 8 andFigure 7`. There are no appropriate spaces.

2. It seems that the authors misunderstand the concept of the target in jailbreak attacks. The target refers to the desired output that the attacker aims to elicit from the model, not the model’s actual output. However, in line 122, the authors describe the target as the malicious output of the attack, which is conceptually incorrect.

**Questions:**

1. I am curious about the sentence in line 165: `In TextGrad (Yuksekgonul et al., 2025), we introduce derivatives and gradients as analogies to characterize textual feedback in LLMs.` What do the authors mean by `we introduce ...`?
2. Could the authors provide details on the computational efficiency of the proposed method? e.g., average number of iterations and the average runtime.
3. Could the authors provide some case studies that illustrate the changes that occur across the iterations?

---

### Official Review · Reviewer_ukQt · 2025-11-01

**Soundness:** 2
**Presentation:** 1
**Contribution:** 2
**Rating:** 2
**Confidence:** 4

**Summary:**

The paper proposes a new research task on Omni-MLLM jailbreak attacks by simultaneously targeting three modalities. The paper validates the jailbreak threat of such kinds of MLLMs.

**Strengths:**

1. The paper focuses on a new research question on jailbreak attacks of Omni-MLLMs.
2. The proposed method achieves good attack performance on the crafted dataset.

**Weaknesses:**

1. Clarity of writing. If my understanding is correct, the paper uses the “textual-gradient” concept to simulate the chain rule for effective attacks, but the actual procedure is: (i) query an existing LLM for improvement suggestions; (ii) apply those suggestions to the three modalities and test the attack; (iii) repeat (i)–(ii) until it works. Here, “textual gradient” is a natural-language critic feedback based on the current prompt. Mapping this process to the chain rule is unnecessary, since many LLM jailbreak works [1,2,3] have already tried essentially the same loop. This also raises a key question: what is the core contribution beyond existing black-box LLM jailbreak [1,2,3] attacks? Apart from applying feedback to three modalities rather than text only, what designs are unique to this task? Overall, the core idea feels somewhat over-packaged; there is no need to force a chain-rule analogy. Section 3.3 has the same issue.

[1] AutoDAN: Generating Stealthy Jailbreak Prompts on Aligned Large Language Models. ICLR 2024

[2] Jailbreaking Black Box Large Language Models in Twenty Queries. Arxiv 2023

[3] GPTFUZZER: Red Teaming Large Language Models with Auto-Generated Jailbreak Prompts. Usenix Security 2024


2. Questions about the experimental setup. In Table 1, how are many MLLM attack methods directly used under the current setting? Given that the paper uses a tri-modal dataset (audio, image, text), how is an image-only attack such as FigStep applied “to ensure fairness”? Did the authors keep the original audio and perturb only the image–text pair, or do something else? This needs clarification.

3. Commercial black-box models. Since the paper adopts a black-box threat model, evaluations on commercial models are necessary to demonstrate practicality and generalization.

**Questions:**

Please refer to the Weaknesses section. Overall, I do not fully follow the authors in casting LLM feedback as a “textual gradient.” Introducing equations ought to clarify the motivation, but in this case, it appears to add unnecessary cognitive burden. I would welcome clarification if I have misunderstood the definition or usage of “textual gradient,” and I am glad to revise my score accordingly.

---

### Official Review · Reviewer_7isR · 2025-11-02

**Soundness:** 3
**Presentation:** 2
**Contribution:** 3
**Rating:** 4
**Confidence:** 3

**Summary:**

The paper uses textgrad methods to optimize inputs for omni-modal LLMs. They empirically show this outperforms a set of strong baselines.

**Strengths:**

The unified attack across multi-modalities is interesting. Empirical results are impressive.

**Weaknesses:**

1. The paper basically adapts textgrad for jailbreak. This simple extention lacks novelty considering other LLM based attack also do it similarly (PAIR, TAP).

2. Missing baselines: PAIR attack can also achieve similar functions by using attacker LLM to refine input prompt based on feedback. I feel this is extention of PAIR attack and empirical comparison evidence would be important.

3. Examples of optimized prompts should be given and highlighted. Qualitative analysis of diversity and stealthiness of examples should be provided.

**Questions:**

Please refer to the weakness part.

---

### Official Review · Reviewer_ok17 · 2025-11-02

**Soundness:** 3
**Presentation:** 2
**Contribution:** 2
**Rating:** 2
**Confidence:** 4

**Summary:**

This paper proposes AdvOmniAgent, a framework for jailbreaking Omni-Modal Large Language Models (Omni-MLLMs) that take text, image, and audio inputs. The approach introduces a dual-phase optimization process that uses LLM-generated feedback (“textual gradients”) to approximate cross-modal gradients and guide multimodal query updates, along with a Feedback-Driven Adaptive Generator Parameter Update (FAGPU) to stabilize optimization. Experiments are conducted across multiple Omni-MLLMs, claiming higher attack success rates than prior baselines such as FigStep, AdvWave, and IDEATOR.

**Strengths:**

1. Tackles a relevant problem as tri-modal LLMs become more widespread.
2. Provides comprehensive experiments and ablations across multiple models.

**Weaknesses:**

1. The paper has limited novelty. The central mechanism of using LLM feedback as “textual gradients” to guide black-box optimization is directly borrowed from TextGrad. The paper even cites this prior work in a self-referential way, making the core idea largely derivative rather than original.
2. The self-citation on line 165 (“In TextGrad … we introduce …”) explicitly reveals authorship of prior work during double-blind review, violating anonymity requirements.
3. The dataset construction is not realistic. AdvBench-Omni is built by adding TTS to jailbreak prompts. This is convenient but may not reflect real tri-modal interactions (e.g., human-recorded speech, non-TTS noise). Robustness to different TTS voices and real recordings is not evaluated.

**Questions:**

How does the proposed dual-phase optimization technically differ from TextGrad beyond adding multimodal prompts?

**Details Of Ethics Concerns:**

The paper explicitly discloses author identity through self-referential citation (“In TextGrad … we introduce …”), violating double-blind review policy.

---

### Note · Authors · 2025-11-13

I have read and agree with the venue's withdrawal policy on behalf of myself and my co-authors.